# Leveraging Emerging Technologies to Expand Accessibility and Improve Precision in Rehabilitation and Exercise for People with Disabilities

**DOI:** 10.3390/ijerph21010079

**Published:** 2024-01-10

**Authors:** T. Bradley Willingham, Julie Stowell, George Collier, Deborah Backus

**Affiliations:** 1Shepherd Center, Virginia C. Crawford Research Institute, Atlanta, GA 30309, USAdeborah.backus@shepherd.org (D.B.); 2Department of Physical Therapy, Georgia State University, Atlanta, GA 30302, USA

**Keywords:** digital health, mHealth, mRehab, mobility impairment, chronic conditions

## Abstract

Physical rehabilitation and exercise training have emerged as promising solutions for improving health, restoring function, and preserving quality of life in populations that face disparate health challenges related to disability. Despite the immense potential for rehabilitation and exercise to help people with disabilities live longer, healthier, and more independent lives, people with disabilities can experience physical, psychosocial, environmental, and economic barriers that limit their ability to participate in rehabilitation, exercise, and other physical activities. Together, these barriers contribute to health inequities in people with disabilities, by disproportionately limiting their ability to participate in health-promoting physical activities, relative to people without disabilities. Therefore, there is great need for research and innovation focusing on the development of strategies to expand accessibility and promote participation in rehabilitation and exercise programs for people with disabilities. Here, we discuss how cutting-edge technologies related to telecommunications, wearables, virtual and augmented reality, artificial intelligence, and cloud computing are providing new opportunities to improve accessibility in rehabilitation and exercise for people with disabilities. In addition, we highlight new frontiers in digital health technology and emerging lines of scientific research that will shape the future of precision care strategies for people with disabilities.

## 1. Introduction

According to recent data from the Centers for Disease Control (CDC), 61 million (26%) adults in the United States are living with disability as defined as “serious difficulty walking, hearing, seeing, concentrating, remembering, or making decision”, with disability from mobility limitations being the most common (14%) [1]. Within this population, people with all types of disabilities experience physical, environmental, structural, social, and economic disadvantages that negatively impact their health and quality of life disproportionately, compared to those without disability. Specifically, when considering health disparities, people with disabilities are three times more likely to have comorbidities such as heart disease, stroke, and diabetes [2,3,4,5] and are more than four-fold more likely to experience mental distress compared to those without disability [6]. Moreover, physical disabilities that impact mobility can limit participation in physical activities and exercise [7,8], and, over time, chronic physical inactivity can lead to physiological deconditioning that further impairs physical function and cardiometabolic health [9,10,11,12,13]. In people with physical disabilities, research has demonstrated that the degree of cardiometabolic disease risk, dependence, anxiety, and depression worsens with the severity of mobility impairment [14,15,16,17,18,19,20]. Thus, there is a profound need for accessible and efficacious treatment strategies to improve physical and mental health in people with disabilities, particularly in those with mobility limitations.

Physical rehabilitation and exercise training have emerged as promising solutions to improving health, restoring function, and preserving quality of life in populations that face disparate health challenges related to disability [3,5,21,22,23,24,25,26,27,28,29,30,31,32,33,34,35,36,37]. Specifically, clinical research studies have shown that rehabilitation and exercise programs provide an opportunity to simultaneously target multiple physiological systems and drive improvements in neural function, cardiopulmonary and metabolic capacity, muscle strength, endurance, flexibility, and vascular health [3,5,21,22,23,24,25,26,27,28,29,30,31,32,33,34,35,36,37] while also reducing mental distress and attenuating depression and anxiety [14,15,16,17,18,19,20] in these populations. Despite the enormous potential for rehabilitation and exercise to help people with disabilities live longer, healthier, and more independent lives, people with disabilities can experience physical, psychosocial, environmental, and economic barriers that limit their ability to participate in rehabilitation, exercise, and other health-promoting physical activities [38,39,40,41,42,43,44] (Figure 1). Here, we discuss how emerging technologies may provide new opportunities to overcome such barriers and establish more accessible and effective approaches to rehabilitation and exercise for people with disabilities. 

## 2. Telecommunications

Participation in physical rehabilitation and exercise often requires access to specific environments, specialized equipment, and professional instruction, which may not be safe, accessible, or relevant for people with disabilities and which can directly limit their ability to participate in such activities. For example, facilities that support rehabilitation and exercise such as clinics, gyms, parks, and trials are often physically inaccessible to people with disabilities who have mobility limitations, and inclement weather can further limit the utility of even the most accessible facilities [45,46,47,48,49,50,51]. In addition to physical barriers, public gyms and crowded spaces may also contribute to intrapersonal barriers, as people with disabilities commonly report barriers associated with self-efficacy, perception of risks, and fear of embarrassment [38,39,40,51,52,53]. Moreover, people with disabilities may also experience barriers associated with transportation and scheduling that negatively impact their ability to attend scheduled rehabilitation and exercise activities at specific locations. Outside of clinically-supervised rehabilitation programs, public-facing fitness organizations and exercise classes often lack expertise, which may limit the availability of safe instruction that is tailored to the needs and concerns of people with disabilities [45,46,47,54,55,56]. Together, these barriers contribute to health inequities in people with disabilities by disproportionately limiting their ability to participate in health-promoting physical activities such rehabilitation and exercise, relative to people without disabilities. 

Tele-health has emerged as a promising solution to expanding access to quality care for individuals that are geographically, physically, cognitively, and economically disadvantaged [57,58,59,60]. “Tele-health” refers to strategies that leverage telecommunications technologies such as telephone, email, text, and the internet to provide healthcare services to individuals in remote locations outside the clinic [57,61,62,63,64,65]. Tele-health interventions can overcome many of the barriers related to facility access, scheduling, and transportation by offering a more convenient and economical solution that can be performed in one’s home or in an accessible location of one’s choosing [57,58,61,66,67,68,69]. As highlighted in this section, rehabilitation and exercise professionals have also adopted tele-health strategies to specifically address the accessibility issues faced by people with disabilities.

### 2.1. Tele-Rehabilitation

The need to deliver home-based rehabilitation programs has been a longstanding challenge for patients and clinicians. In addition to overcoming many of the barriers patients encounter when accessing rehabilitation services, home-based rehabilitation interventions also address fundamental treatment gaps that may ultimately improve outcomes for patients. For example, in response to a research survey administered to over 500 rehabilitation professionals, clinicians reported that 70% of their patients with disabilities required additional therapy after discharge, and more than 55% required additional therapy between in-clinic visits [70]. In this same cohort, 95% of the respondents indicated that tele-rehabilitation may provide an opportunity to address the need for additional treatment sessions [70]. Over the past 25 years, the utility and efficacy of tele-rehabilitation has been evaluated in several different clinical populations [71,72,73,74,75,76,77], and, according to the National Library of Medicine’s PubMed database, the number of peer-reviewed research articles featuring tele-rehabilitation in the title alone has increased more than 10-fold in the past decade. Empirical evidence from tele-rehabilitation research studies suggest that the use of these tele-health strategies may provide an accessible and effective option for deploying rehabilitation and exercise interventions for people with disabilities [57,61,64,65,69,78,79,80]. Recent systematic and thematic reviews of this body of literature have concluded that the available evidence demonstrates that tele-rehabilitation interventions are effective at improving physical function, cardiometabolic health, and quality of life in people with disabilities associated with neurologic, musculoskeletal, and cardiovascular disorders [57,59,61,64,65,69,79,80,81,82]. For example, studies evaluating the impact of tele-rehabilitation interventions in people with mobility impairment associated with neurologic conditions (stroke, multiple sclerosis, and spinal injury/disease) have found remote intervention strategies to be as effective as standard, in-person rehabilitation at improving measures of physical function and mobility [57,59,61,62,63,72,76,81,83,84,85]. 

Historically, home-based rehabilitation programs have been limited to using detailed written instructions and illustrations, before evolving into video-based instructions delivered through videotape or digital video disk [57,58,59,61,64,69,72,84]. More recently, the advent of high-speed internet, massive data storage capacity, and affordable digital videography have revolutionized in-home rehabilitation programming, through the emergence of web-based tele-rehabilitation [86,87,88,89]. Specifically, these advancements in telecommunications technology offer a more versatile approach to tele-exercise, which allows clinicians to efficiently create and upload large libraries of custom video content that can be accessed by users from any device with internet connectivity (i.e., tablet, smart phone, etc.). Furthermore, the increasing affordability and quality of real-time video streaming and video conferencing has also expanded the capabilities of tele-rehabilitation by enabling the live streaming of video content and allowing patients to interact with their healthcare providers through real-time two-way video conferencing in a secure, interactive environment [90,91]. While tele-rehabilitation delivered through various methods has demonstrated immense potential to make a substantial impact on health and function in people with disabilities, the available evidence also suggests that the effectiveness of tele-rehabilitation interventions may be limited by poor adherence and a lack of clinical supervision [57,59,61,64,65,69,75,79,80,81,82,92]. Undoubtedly, corrective and encouraging feedback is critical to the safety and effectiveness of rehabilitation, but this is difficult to provide during tele-rehabilitation sessions where direct clinical supervision is limited or absent altogether. Thus, future work is needed to address the limitations of tele-rehabilitation interventions and identify ways to improve sustained engagement and expand clinical evaluation into remote settings. 

### 2.2. Tele-Exercise

In addition to clinician-guided rehabilitation interventions, mounting evidence has also demonstrated that regimented exercise can improve physical function and overall health in people with disabilities by increasing cardiopulmonary and metabolic capacity, muscle strength, coordination, and vascular health [21,33,36,93,94,95,96,97,98,99,100,101,102,103,104,105,106] while also reducing mental distress and attenuating depression and anxiety [2,25,107,108,109,110] in these populations. Yet, people with, and without, disabilities encounter a multitude of intra- and interpersonal barriers to participating in exercise programs that involve professional supervision, specialized facilities, or public spaces. To this end, telecommunications technologies have also been rapidly adopted by exercise professionals, and within the last half decade, the entire fitness industry has rapidly evolved into a hybrid ecosystem, as the popularity of online fitness programming has grown exponentially [111,112,113]. 

According to a survey of over 4000 fitness professionals conducted by the American College of Sport Medicine, online fitness climbed to the #1 most popular fitness trend in the fitness industry in 2021 [111,112]. However, the preponderance of public-facing tele-exercise platforms, such as the commercial tele-exercise programs designed by Peloton [114] and Tonal [115], provide content and user interfaces that are primarily curated for the general population and may not be suitable to the needs of people living with disabilities. Therefore, it may be difficult for someone living with a disability to identify commercially available exercise content that provides a safe and accessible effective tele-exercise platform that will help them attain their goals [116]. Furthermore, studies evaluating tele-exercise programs in clinical populations have consistently found that participants reported limited social engagement as a primary limitation to tele-exercise strategies [75,117,118,119]. Thus, while telecommunications technologies have immense potential to address health equity in people with disabilities, there is a need for more accessible and efficacious approaches to tele-exercise programs that are socially engaging and tailored to the needs of people with disabilities. 

### 2.3. Expanding Tele-Health Solutions

Solutions that leverage emerging technologies to expand clinical evaluation into remote settings and integrate social connectivity into tele-health platforms will be a vital component in the future of tele-rehabilitation and tele-exercise. As discussed in detail below, a growing body of research is starting to reveal the many ways remote monitoring tools, such as wearable sensors and digital survey technologies, may be used to evaluate physical function and physiology in remote settings (see “remote monitoring”) [120,121,122]. In addition, some companies are exploring approaches to remote monitoring that directly integrate clinical evaluation into the tele-rehabilitation platform itself. For example, Kaia Health [123], is a tele-rehabilitation platform that leverages computer vision technology to map a patient’s biomechanics based on two-dimensional video data captured from a smartphone camera to evaluate a patient’s movements while they are viewing instructional video content through a mobile app. The software then provides automated, corrective feedback in real-time during their tele-rehabilitation activities, while also providing the opportunity for asynchronous clinician-mediated feedback, based on recordings. Other technologies, such as chatbots, are also being explored as a method to improve adherence in tele-rehabilitation by providing support in the form of tailored feedback and encouragement [124,125,126]. Specifically, some studies have shown how automated, SMS-mediated chatbot interactions can provide patients with information and encouragement, which may improve adherence to an assigned tele-rehabilitation [124,125,126]. Ultimately, establishing sustainable and efficacious approaches to tele-rehabilitation will require the integration of multiple different technologies such as those that data from wearable sensors, computer vision systems, and other remote monitoring technologies are used to drive automated feedback from by chatbots and other forms of digital communication. 

The emergence of creator-driven and socially interactive digital health platforms may also provide novel solutions to address the limitations of modern tele-rehabilitation and tele-exercise programs [116,127]. Online tele-health platforms that emphasize social connectivity and allow users to directly communicate with each other and instructors could overcome limitations to tele-exercise associated with the lack of social interactions [2,52,127]. Moreover, tele-health platforms that provide an opportunity for rehabilitation and exercise professionals to organically publish original content may also establish diverse libraries of instructional content that are inclusive of the needs and goals of people with disabilities. For example, Burnalong [128], a digital wellness platform, offers an online community where credentialed creators can post video content such as exercise classes and programs. This model attracts hundreds of different creators with a wide range of expertise who generate a diverse library of tele-exercise video content that is tailored to the needs of various groups, including those with disabilities. The platform also allows users to interact with each other and the exercise instructors as well, through discussion forums and two-way video streams. Therefore, users with disabilities can find professionally curated content that is inclusive of their goals, ability level, and preferences, while also engaging with other individuals who may have similar lived experiences or health goals. Together, innovative tele-health platforms and remote monitoring technologies are starting to bridge the gap between the individual, clinic, and the communities of people living with disabilities. While these solutions have immense potential to collectively address many of the limitations of tele-rehabilitation and tele-exercise, future research is warranted to determine how, and to what extent, applications of remote monitoring and socially engaging platforms may improve adherence and outcomes for people with disabilities. 

## 3. Remote Monitoring

While tele-rehabilitation and tele-exercise have emerged as potential approaches to overcoming many environmental and economic barriers to participating in health-promoting physical activities [57,61,62,63,64,65,80,118], remote interventions are limited by their inability to employ contemporary evaluation strategies and adaptive equipment that may greatly enhance the safety, accessibility, and precision of an intervention. For example, the deployment of safe and effective rehabilitation and exercise interventions requires individualized evaluation strategies that can identify changes in physical function and physiology over time, so that activity intensity and type can be adjusted as an individual adapts [97,129,130]. Yet, key clinical outcome measures, such as walking tests and cardiopulmonary assessments [131,132,133,134,135,136,137], require direct clinical supervision or costly, immobile instruments that cannot be used in remote settings outside the clinic, and therefore, it is difficult to conduct a rigorous evaluation of physical function and physiology of individuals participating in remote interventions, which may negatively impact risk and efficacy. 

### 3.1. Wearable Sensors

Recent advancements in wearable sensor technologies have expanded our capacity to monitor various aspects of human movement and physiology [120,138,139,140,141,142,143,144,145,146,147,148]. Wireless, noninvasive instruments such as accelerometers and optical sensors can now be combined into a single, wearable electronic device, which measures multiple physiological systems in remote locations without any, or with little, clinical oversight required [120,121,122,139,149,150,151,152,153,154,155,156]. By integrating these measurement devices with contemporary system-on-a-chip technology, high-capacity lithium batteries, and Bluetooth connectivity, modern wearable devices can now generate and transmit continuous, high-resolution measurements for days or weeks at a time. Certainly, there is a great deal of consumer interest in wearables, and companies such as Fitbit [157,158,159], Apple [159,160], Garmin, and Whoop [161,162] provide commercially available, user-friendly products such as watches and wristbands that enable users to track their physical activity, cardiac function, sleep, and other aspects of their health, without any technical training required. However, the potential of wearables is also recognized by clinicians and researchers as well, and there has been a rapid expansion in research related to wearable devices over the past decade [120,138,139,140,141,142,143,144,145,146,147,148,163,164]. Moreover, similar to the trends observed in tele-rehabilitation research, the number of peer-reviewed research articles featuring “wearables”, in the title alone, has increased more than 12-fold in the past decade, according to the NLM’s PubMed database. These studies have demonstrated a wide range of potential use cases in which wearable sensors may overcome barriers to accessing evidence-based care by expanding clinical evaluation into remote settings; they provide new opportunities to remotely deliver rehabilitative interventions that require specialized equipment that was historically only available in the clinic [122,149,156,165,166,167,168,169]. 

Because wearable devices can incorporate many different instruments and may be integrated into many different types of wearable accessories, there are a wide range of potential clinical use cases for wearables. Within the wearable device ecosystem, sensors that use inertial measurement units (IMUs) to evaluate movement are amongst the common [120,121,122,141,143,170]. IMUs are noninvasive, electronic sensors that use accelerometers and gyroscopes to simultaneously measure linear acceleration and angular rotation. A wide range of different movements and biomechanics can be measured by pairing wearable IMU sensors with straps and adhesive to secure the devices to the wrist, ankle, hip, or other desired anatomical location. IMU sensors worn at different anatomical locations can provide reliable measures of general daily activity (i.e., steps) [143,151,171,172,173,174,175,176,177,178,179], gait metrics [145,180], spasticity [170,181], and specific upper- and lower-extremity movements [149,182,183,184,185,186,187,188]. Several research studies have used wearable IMUs to reveal important, multidirectional interactions among non-structured physical activity, rehabilitation, and exercise [148,151,179,189,190,191]. Specifically, studies have shown that reductions in daily, non-structured physical activities are related to increased risk of cardiometabolic disease and reduced mobility in people with disabilities, and that participation in rehabilitation and exercise can improve health outcomes, both directly and indirectly, through associated increases in physical activity.

In addition to research instruments, some commercially available devices also leverage IMUs to facilitate participation in home-based exercise activities. For example, Flint Rehab [192] has developed a system called FitMi [193,194,195] that integrates wearable IMU sensors with an interactive software program to monitor and provide feedback regarding movement quantity and quality during participation in tele-exercise activities. Systems such as the FitMi are creating opportunities for people with disabilities to engage in tailored exercise programs that are guided by real-time feedback based on their precise movements. Together, this work demonstrates how wearable IMU sensors may provide an opportunity to remotely evaluate the quantity and type of physical activities associated with tele-rehabilitation and exercise programs.

Measurements of walking function and gait mechanics are critical to evaluating the progression of physical disability and the impact of therapeutic strategies for people with mobility impairments [196,197]. While IMUs sensors embedded in wrist straps and smartphones can provide valuable information regarding physical activity and steps, these devices are limited in their capacity to evaluate lower-extremity biomechanics-associated gait kinetics. To this end, recent developments in wearable sensor technologies have started to incorporate small, wireless pressure sensors into footwear to capture lower-extremity pressure patterns generated by foot placement during gait cycles [198,199,200,201,202,203]. For example, Sensoria Health [204] has developed textile pressure-sensor-infused socks that can evaluate spatiotemporal parameters of gait associated with walking and running. Assessments of gait derived from smart socks have been validated by multiple independent research studies, which have found the sock sensor to be as, or more, accurate than gold-standard gait lab systems [202,203,205]. These preliminary studies have demonstrated exciting potential for smart socks to provide reliable measurements of gait in community-based settings, without the need for direct, clinical supervision. In addition to measuring specific spatiotemporal gait metrics associated with clinical disability, smart footwear may also provide a direct measurement of daily step count that is more inclusive to people with disabilities. The reliability of step counts derived from contemporary wrist-worn sensors is significantly reduced in clinical populations with lower gait speed, and wrist-worn trackers are unable to accurately count steps in populations that use assistive devices, such as walkers, which limit upper-extremity mobility during walking [190,191,206]. Establishing an outcome measure that can accurately evaluate daily steps in people with more severe mobility impairments will create new opportunities to explore the clinical importance of daily physical activity in people with disabilities, and will advance our knowledge of how various treatment strategies impact everyday life. 

In addition to measuring human movement and biomechanics, wearable devices are also commonly used to monitor heart rate, which may be useful when delivering tele-rehabilitation and tele-exercise programs. For example, wearable heart-rate monitors can allow clinicians to monitor cardiac activity during participation in tele-rehabilitation, which may be critical to maintaining safety and identifying problems in patients with cardiac irregularities/impairments [66,139,154,156,179]. Furthermore, evidence-based guidelines for exercise in people with disabilities have been established, based on research showing that the intensity of exercise should be optimized to generate a specific physiological stimulus that can safely and effectively drive improvements in physical function, fatigue, and cardiometabolic health [97,129,130,207]. Thus, wearable sensors may also address this need by providing constant measures of heart rate that can be used to inform the calibration and progression of exercise intensity [107,129,208,209]. However, in populations with neurological conditions, autonomic dysregulation and/or the use of certain medications can disrupt the cardiovascular response to exercise [97,210,211,212,213]. In such cases, heart-rate measurements may not accurately reflect the intensity of physical exertion, and some individuals may not be able to achieve the target heart rate recommended by exercise guidelines. While exercise guidelines for people with neurological conditions can also be achieved by exercising, according to how hard someone perceives they are working [97,214], measuring perceived exertion [215,216] requires tools that can capture subjective, self-reported measures of individual experience.

### 3.2. Ecological Momentary Assessments (EMAs)

Advancements in mobile software technology have led to the development of digitally deployed surveys and questionnaires that can capture self-reported, ecological momentary assessments (EMAs) of symptom perception and mental health outcomes. EMAs evaluate perceptions and behaviors in real time in real-world environments [217,218,219,220,221] and have been used across a broad range of psychological research and clinical practice to evaluate psychological responses to experiences and mental health [217,218,219,220,222,223,224,225]. Software applications, such as RedCap [226] and ExamMed [227], allow researchers and clinicians to build custom EMAs that are sent directly to individuals through their email, text message, or mobile application. Because EMAs operate by prompting users to record their perceptions and thoughts in real time, in real-world environments, this approach minimizes error, such as recall bias, which might influence clinical assessment that requires recall over the course of several days or weeks. In addition, EMAs allows researchers and clinicians to better understand factors that influence symptom perception and mental health by evaluating events that occur in close temporal proximity (before and after) to EMA measurements [219,220,222,223]. Indeed, recent studies have demonstrated how information derived from EMAs can be integrated with activity and geolocation data from wearable sensors to identify key determinants of behavior in community settings, and these studies have started to reveal novel relationships between mood and free-living physical activity [217,219,220,222,228]. While similar evaluation strategies have great potential to inform rehabilitation and exercise strategies, there are currently limited methodologies established to integrate EMAs with multiplex data from multiple instruments in modern wearable sensors. 

Integrated remote monitoring systems that combine quantitative data from wearable sensors with subjective data from EMAs could provide new opportunities to explore how physical and mental health are influenced by disease, disability, and related interventions in real-world environments. For example, people with disabilities often report fatigue and pain as significant barriers to participating in rehabilitation and exercise activities [4,229,230,231]. Fatigue and pain can be extremely variable from day to day, and even within a single day [232,233,234,235,236,237], and a few contemporary studies have used daily surveys and remote activity trackers to show that momentary feelings of fatigue and pain can have an immediate, negative impact on daily physical activity [232,236,237]. Yet, the preponderance of studies evaluating fatigue in people with disabilities have relied on questionnaires that ask individuals to provide a summative rating of the fatigue they have experienced over the past 7–28 days [238,239]. Therefore, it is difficult to delineate how fatigue impacts participation in physical activities on a day-to-day basis. Consequently, there is a lack of empirical scientific evidence regarding how daily fluctuations in fatigue and pain might acutely influence exercise participation in people with disabilities, and it is unclear whether long-term adaptations from exercise impact the relationship between fatigue, pain, and non-exercise physical activities in real-world settings in this population. Future studies implementing remote monitoring strategies that integrate physical activity and physiology metrics with perceived outcomes may help identify acute and chronic interactions among rehabilitation and exercise, fatigue, and pain in people with disabilities on a day-to-day basis in real-world settings. Such work could help clinicians develop and deploy more sustainable rehabilitation and exercise strategies for people with disabilities who experience elevated levels of fatigue and pain.

## 4. Digital Environments and Gaming

Safety, engagement, and adherence are critical components for delivering sustainable and effective rehabilitation and exercise programs. Undoubtedly, rehabilitation and exercise often require activities that are associated with a moderate risk of falling or injury, and people with deficits in motor control, balance, and cognition may be at greater risk during participation in such activities [97,130,240,241,242]. While clinicians can help mitigate risks by providing detailed instructions and physical assistance, remote strategies lack clinical oversight, which may lead to additional challenges related to safety and engagement and that lead to reduced adherence. Moreover, some intentional aspects of rehabilitation and exercise activities may present inherent safety concerns that persist even with professional oversight. For example, rehabilitation strategies that incorporate external stimuli and focus on walking around objects have been shown to improve walking function in people with disabilities [243], but such obstacles may increase the risk of an adverse event such as a fall and/or injury. The level of risk and an individual’s perception of safety during participation in rehabilitation and exercise programs may also contribute to hesitancy, and studies have shown that people with disabilities report safety concerns as a primary factor that contributes to low adherence [39,41,42,52,53]. Thus, maintaining environments that have reduced risk of adverse events and help patients feel safe during their participation in rehabilitation and exercise activities may also improve adherence and engagement. 

In addition to safety, people with disabilities also commonly report a lack of motivation as another factor that may negatively impact adherence to rehabilitation and exercise programs [39,41,42,52,53]. During supervised rehabilitation and exercise activities, clinicians play a key role in keeping patients engaged in their activities by providing real-time instructive feedback and incorporating simple, goal-oriented tasks such as games. Specifically, studies have shown that rehabilitation strategies that incorporate feedback, games, and elements of entertainment may increase the intensity at which a specific activity is performed [244,245,246,247]. In this section, we highlight emerging technologies that are providing new opportunities to improve safety, engagement, and adherence in rehabilitation and exercise programs for people with disabilities. 

### 4.1. Virtual and Augmented Reality

Virtual-reality (VR) and augmented-reality (AR) technologies provide an opportunity to address challenges associated with safety, engagement, and adherence in rehabilitation and exercise for people with disabilities [77,245,248,249,250]. VR and AR technologies use computer-generated imagery to create immersive and interactive digital environments. Specifically, VR establishes a digital environment that reflects real or fictional places, whereas AR is the real world viewed through a device, adding computer-generated objects or effects into one’s domain. Both VR and AR offer the opportunity to safely simulate walking or other daily activities, in various digital environments that may include obstacles or everyday scenarios, with a reduced risk of harm [249,251,252,253,254]. Moreover, VR and AR provide a method of integrating fantastical situations and gaming into rehabilitation and exercise activities, to make them more enjoyable and/or entertaining [253,254,255]. 

While multi-sensory digital experiences date back to the 1990s, recent developments in computer-generated imagery and microprocessors have expanded the quality and mobility of VR/AR technologies, and have consequently expanded their potential applications for rehabilitation and exercise [77,249,250,251,256,257]. Specifically, a growing body of research has demonstrated the safety and efficacy of integrating VR technologies into rehabilitation care plans in several rehabilitation populations, including stroke, multiple sclerosis, rheumatoid arthritis, fibromyalgia, and Parkinson’s disease, among others [77,249,250,251,256,257]. The use of VR technologies as a rehabilitation tool has been shown to improve mobility and function, decrease pain, and improve quality of life [77,249,250,251,256,257]. Moreover, VR may also decrease the burden on caregivers, while raising compliance, motivation, and adherence [250,253], and research indicates that skills learned in virtual environments can translate to the real world [250,253]. Regarding AR, some evidence suggests that AR-based interventions may improve physical function and reduce risk of fall when implemented in conventional physical therapy practice for people with disabilities [245,248]. However, there are a limited number of research studies investigating AR interventions, and the degree to which the integration of AR technology may contribute to additional improvements in outcomes, over and beyond traditional approaches to rehabilitation, remains unclear. 

Contemporary VR and AR devices have many attributes that allow them to be used across a variety of rehabilitation and exercise strategies. For example, VR and AR can be combined with numerous accessories, such as omnidirectional treadmills and haptic suits, which incorporate physical activity and feedback with the virtual or augmented experience [258,259,260]. VR and AR devices are also becoming more mobile and integrated into accessories that enable everyday use, such as wireless eyewear, smartphones, and heads-up displays (HUDs) [261,262]. VR systems have also become increasingly accessible for individuals who use wheelchairs or have limited mobility. Specifically, some VR programs now offer seated experiences, making it suitable for those who prefer or require a stationary position. VR developers are also incorporating accessibility features, including adaptive controllers, customizable user interfaces, subtitles, and adjustable avatars.

Systematic reviews evaluating the usability of semi-immersive and fully immersive VR systems have reported strong usability in people with a wide range of neuromotor impairments, particularly when examining user ratings regarding ease-of-use and learnability [263]. Notwithstanding, VR and AR have limitations that reduce their clinical utility and require further development to establish more accessible and efficacious virtual environments for use in rehabilitation and exercise. For example, VR and AR use in rehabilitation and exercise can be associated with motion sickness, loss of human connections, safety issues for those with limited mobility, and cost [264,265,266] which may be particularly problematic for people with disabilities. Fortunately, recent development work has focused on overcoming many of the limitations of VR and AR. Companies such as Meta [267] and Apple [268] are in the process of developing more connected virtual environments, to enable users to remotely interact and exchange information in the metaverse [269,270]. Such innovations have the potential to open a world of possibilities for people with disabilities who have barriers to transportation or limited ability to spend long periods of time away from home. Google has also created Google Cardboard, a smartphone assembly for digital environments made of cardboard that has potential to reduce costs of VR and AR technology [271]. Furthermore, emerging products, such as Teslasuit [272] are now offering full-body haptic experiences that provide comprehensive feedback to multiple anatomical locations. The multidirectional feedback received from the suit is suggested to increase the body’s response to input from VR and/or AR, thus potentially stimulating neural pathways and propagating stronger synaptic connections [273] which are important targets for neurorehabilitation. Collectively, these advancements will lead to more realistic, affordable, and socially connected digital environments, which will establish more accessible and efficacious VR and AR solutions for rehabilitation and exercise. 

### 4.2. Gaming

Digital gaming technologies have been around for over 50 years, and over the past decade, the gamification of rehabilitation and exercise has emerged as a promising strategy to improve engagement and adherence [167,244,248,264,274,275]. Gamification refers to the “use of game design elements in non-game contexts” [275]. In the context of rehabilitation and exercise, gamification may involve the use of quests, points, levels, leaderboards, or badges to guide and encourage users through a series of physical activities [167,244,248,264,274,275]. Users frequently report that gaming elements are fun, exciting, and reduces the monotony that patients may experience during their plan of care, which can increase adherence and performance during rehabilitation and exercise activities [167,244,248,264,274,275]. Moreover, the stimuli of gaming elements themselves may also initiate changes in specific neural adaptations [264]. Specifically, gaming has been shown to release dopamine with each achievement, facilitating neuroplasticity and potentiation of synaptic connections [244], and game-based therapies involve cognitive and motor components that recruit specific neurons of interest, leading to the formation of new neural networks [266].

The benefits of integrating non-immersive digital gaming into rehabilitation and exercise have been evaluated in different clinical populations [276,277], and there are several studies demonstrating how gaming platforms such as Microsoft Kinect and Nintendo Wii can improve physical function and overall wellbeing in people with disabilities [247,278,279,280,281,282]. Advancements in gaming are also offering new opportunities for people to engage in collaborative gaming activities through online, multiplayer games. Online games can overcome social barriers for people with disabilities, while also allowing people with mobility limitations to engage and compete in gamified activities that may be otherwise physically inaccessible [283,284,285]. More recently, clinicians and researchers have also started to explore how gamification of rehabilitation and exercise can be delivered using VR technologies [286,287,288]. Strategies that guide individuals through movements using gamified rehabilitation and exercise activities performed in immersive VR environments offer an approach that provides all of the benefits of gamification in addition to the enhanced user experience and psychophysiological stimuli generated by VR. Although future research is needed to determine the best practices for combining VR and gaming in a rehabilitation context, this integrated strategy has the potential to provide a safe, accessible, and motivating approach for people with disabilities, while also generating robust stimuli to drive adaptations within multiple physiological systems [248,255,274,278]. 

Taken together, the collective evidence demonstrates how telecommunications, wearables, VR, AR, and gaming may be used either individually, or in combination, to overcome many barriers people with disabilities face regarding rehabilitation and exercise. However, choosing the most effective combination of technological solutions for each individual is dependent on several factors related to personal goals, abilities, and preferences that are shaped by a myriad of physiological, psychological, sociological, and economical factors. In the next section, we highlight new frontiers in precision care strategies that are helping clinicians make data-driven decisions when it comes to selecting the most effective and efficient rehabilitation and exercise strategies for people with disabilities. Yet, the overall value of integrating VR and AR technologies into rehabilitation remains somewhat unclear. Future research focused on demonstrating how VR and AR may enhance outcomes and cost effectiveness, over and beyond conventional care strategies, may facilitate the translation of these technologies into clinical practice.

## 5. Promoting Accessibility with Co-Development

While digital health technology holds significant promise in increasing accessibility, adoption may not be universally seamless across diverse user groups, and it is crucial to recognize potential barriers that individuals with disabilities may encounter when utilizing technological devices and digital health services. For instance, individuals with mobility impairments or motor control challenges may encounter challenges when physically interacting with small buttons, straps, touchscreens, or complex clasp mechanisms that are often associated with wearable sensors and mobile telecommunications devices. Sensory impairments among individuals with visual or hearing deficits may also limit the usability of digital health technologies that do not offer alternative modalities for information presentation. In addition to physical barriers, people with disabilities may also experience cognitive impairments that make it difficult to navigate complex digital interfaces or manage complex protocols associated with operating and maintaining digital health devices and platforms. Thus, it is critical that future technological innovation efforts are informed by preemptive and parallel research agendas that are focused on understanding the potential barriers to implementing digital health technologies within the population of individuals with disabilities. Co-innovation strategies directly involve people with disabilities in the technological innovation process by conducting a series of focus groups, surveys, and pilot studies to identify the needs, barriers, and preferences of the intended user [117,289,290,291] and systematically evaluate the feasibility, usability, and safety of technologies. Through this iterative process, co-development strategies allow developers to design and refine digital health solutions that are not only effective, but also inclusive and accessible to individuals with disabilities (Figure 2). 

## 6. Precision Care

While substantial progress has been made in establishing evidence-based guidelines for rehabilitation and exercise in people with disability [97,130,207], standardized guidelines only provide general recommendations, which are not directly tailored to the needs, abilities, and goals of each individual. Undoubtedly, people with disabilities can have a wide variety of mobility limitations and physiological disorders that differentially impact how they participate in, and how their body responds to, distinct types of interventions. Thus, there is increasing awareness in the field of physical medicine, rehabilitation, and exercise physiology surrounding the need for individualized, data-driven treatment strategies that are specifically tailored to an individual’s needs, goals, and abilities. To this end, the concept of precision care (i.e., precision medicine/rehabilitation) has emerged as an alternative approach to the generalized, broad application of evidence-based guidelines [292,293,294,295,296]. Precision rehabilitation leverages large, multiplex datasets to establish personalized, data-driven treatment strategies that are optimized to drive outcomes for individuals (Figure 3). For example, one approach to precision rehabilitation is to establish patient subtyping systems, in which subgroups of patients with the same diagnosis are identified and differentially assigned to specific treatments based on their integrated health profile [292,293,294,295,296]. In addition, precision rehabilitation strategies may also support more dynamic approaches to care that incorporate new health data as it is generated to make real-time adjustments to treatment strategies [165,292,294]. In addition to addressing barriers related to the lack of individualized care for people with disabilities, precision rehabilitation treatment strategies may also lead to more efficient and economical clinical practices, by creating data-driven models of care that concentrate resources on the specific rehabilitation services that are most likely to make an impact for each patient. Notwithstanding, precision rehabilitation is a relatively new concept, and, to date, only 13 peer-reviewed articles have been published with “precision rehabilitation” in the title. Substantial research and innovation efforts will be required in this area to establish pathology-specific best practices and develop robust technical solutions that can support the implementation of precision rehabilitation at scale. 

A primary limitation to precision rehabilitation strategies is the data collection, as well as the processing and analysis bottleneck that is created by establishing the large, multiplex datasets that result from integrating several sources of health information [292,297]. For example, precision rehabilitation strategies such as patient subtyping must be able to determine which types of treatments generate the most favorable outcomes for specific patient subtypes. Because patient outcomes depend on several physiological, functional, social, and environmental factors, generating accurate predictive models requires valid data and robust analytical approaches that can perform complex, iterative analyses to identify patterns and predictors within large, multiplex datasets that can cover several years of time. Therefore, the practical implementation of precision rehabilitation necessitates the development of automated, user-friendly data collection and analytical systems that can deliver actionable information to clinicians without the need for hours of manual or supervised data collection and analysis [292,297,298,299,300]. 

### Artificial Intelligence and Cloud Computing

Emerging evidence is starting to reveal how applications of artificial intelligence (AI) may be used to automate analyses and identify patterns within large, integrated health datasets, which may be difficult for humans to otherwise detect. AI broadly refers to a form of computer science where algorithms are trained to perform complex tasks, such as pattern recognition. For example, applications of AI have made substantial contributions to the fields of oncology, neurology, and immunopathology, by revolutionizing, and automating, our capacity to rapidly and accurately analyze high-resolution images generated by medical imaging instruments [301,302,303,304,305,306,307]. Historically, biomedical imaging relied on manual analyses by physicians and scientists to identify and quantify biomarkers within large images containing many different relevant and irrelevant features. AI transformed this practice by using algorithm-based image analysis to drastically increase the rate, scale, and sensitivity of biomarker detection [301,307,308,309,310].

Similar to images, integrated health datasets used in precision rehabilitation strategies also include massive amounts of information that may or may not be relevant to an individuals’ prognosis. Therefore, it would follow that applications of AI also have the potential to drastically increase the rate, scale, and overall capacity to extract clinically meaningful information from datasets consisting of hundreds of metrics derived from multiple different sources. For example, recent work has demonstrated how AI algorithms can use multifactorial data derived from electronic medical records (EMRs) to predict hospitalization [311,312], drug efficacy [313], and the development of critical illness [314] in several different clinical populations. One limitation of this approach is that AI predictive models often provide little information about which factors may be driving the outcomes. Indeed, the algorithms used in medical applications of AI consider hundreds of factors, and systematically evaluate interactions and patterns among hundreds of factors including demographics, lab results, co-morbidities, social determinants of health, and medical images, among many others. However, identifying key factors associated with positive or negative outcomes is critical to designing targeted treatment strategies for people with disabilities. To this end, there have been recent efforts to employ “explainable” AI models that systematically identify key factors that are driving the predictive power of the algorithms [315,316]. For example, while it may be helpful to know which patients will adhere to their rehabilitation or exercise program, systems that use explanatory AI may help identify the primary reason patients are, or are not, adhering to protocols, which may inform future intervention strategies.

Although there is increasing interest surrounding the use of AI in precision rehabilitation [292,294,296], the application of AI in rehabilitation is still in the nascent stages of exploration. Several previous studies have demonstrated the utility of using machine learning to forecast outcomes related to walking function, assistive-device use, discharge potential, and falls risk in people with disabilities [315,317,318,319,320,321,322], and such applications of AI can certainly inform the design and management of individualized rehabilitation treatment strategies. Yet, further work is needed to systematically evaluate the safety and efficacy of using AI to direct an individual’s care through processes such as patient subtyping or AI-directed FES calibration [292]. Moreover, existing studies have focused on applying AI to datasets derived from a single source, such as EMRs, wearables, or specialized clinical measurement devices [315,317,318,319,320,321,322]. Undoubtedly, approaches that integrate data from these different sources, and others, would offer a more comprehensive profile of the patient, and may improve the precision of AI-generated predictive models aiming to forecast outcomes or determine the optimal care path for a particular patient subtype. In addition, data integration may also inform dynamic-precision rehabilitation strategies. For example, combining static EMR data with real-time data from remote patient-monitoring systems could provide deep insight regarding the acute and chronic impact of a rehabilitation treatment session if such data is integrated with the treatment schedule or information from tele-rehabilitation/tele-exercise platform [165]. Therefore, in addition to the need for complex, automated analytics, another substantial challenge for precision rehabilitation is the need for a robust technical infrastructure that can rapidly integrate, organize, and store large, multiplex datasets, in a readily accessible and analyzable format. 

Cloud-based data repositories and computing systems have emerged as a promising solution to the data storage and processing demands of precision rehabilitation strategies that aim to leverage big data [323,324,325,326]. For example, EMR systems are often isolated from other sources of health information such as data derived from rehabilitation equipment and remote patient-monitoring systems. Moreover, even the most simplistic wearable devices can generate thousands of data points per day per patient, which can cause data storage and processing difficulties for consumer-level desktop computers. Contemporary cloud systems, such as those offered by Microsoft Azure [327] and Amazon Web Services [328], provide a robust, versatile infrastructure for addressing data integration, storage, and processing needs. Furthermore, the AI strategies highlighted above require immense computational power to support the complex, iterative analytical approaches used to identify patterns and predictors within large, multiplex datasets. While historical medical records contain a wealth of information related to patient outcomes, even the most sophisticated EMR systems lack the analytical tools and AI capacities needed to turn historical outcomes data into predictive algorithms for precision rehabilitation. Contemporary cloud-based computing services also address this need by offering access to nearly unlimited computational power on an as-needed basis. As healthcare technology continues to evolve, the complexity and size of the health data ecosystem will only continue to grow, and thus, data-driven approaches to healthcare also require versatile solutions that rapidly adapt and match the capacity of the continuously evolving data sources. To this end, the capabilities of cloud computing provide the technical infrastructure needed to establish scalable, big-data solutions to support contemporary and future approaches to precision medicine. 

## 7. Conclusions

Emerging digital health technologies have the potential to significantly impact the accessibility and precision of rehabilitation and exercise strategies for people with disabilities. Telecommunications technologies are expanding access to care by providing remote rehabilitation and exercise services to individuals in remote or underserved areas, or to those who are unable to access in-person rehabilitation services, due to mobility or other accessibility barriers. Wearable devices and sensors can be used to monitor a person’s progress, provide real-time feedback, and generate data to inform the progression of rehabilitation and exercise programs. Furthermore, VR and AR offer immersive and interactive digital environments that can be used in rehabilitation and exercise to guide users through specific movements, simulate real-world scenarios, and integrate elements of gaming and entertainment into their training. Advancements in artificial intelligence and cloud computing are also improving rehabilitation and exercise for people with disabilities by enabling data-driven approaches to precision rehabilitation that are directly tailored to the needs and goals of the individual. Overall, emerging technologies have the potential to enhance and expand the capabilities of rehabilitation professionals, and to provide new and innovative ways to help individuals with disabilities or injuries achieve their goals and improve their quality of life. 

## Figures and Tables

**Figure 1 ijerph-21-00079-f001:**
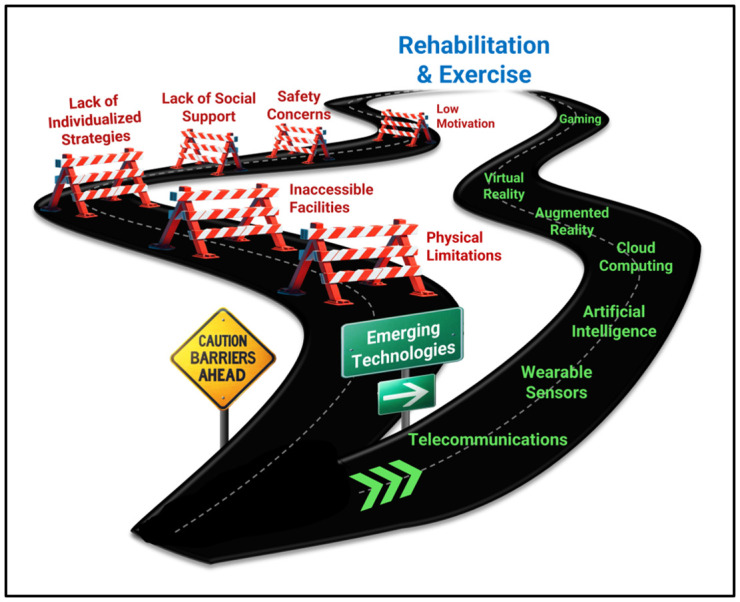
Emerging digital health technologies are providing opportunities to overcome barriers experienced by people with disabilities to participation in rehabilitation and exercise.

**Figure 2 ijerph-21-00079-f002:**
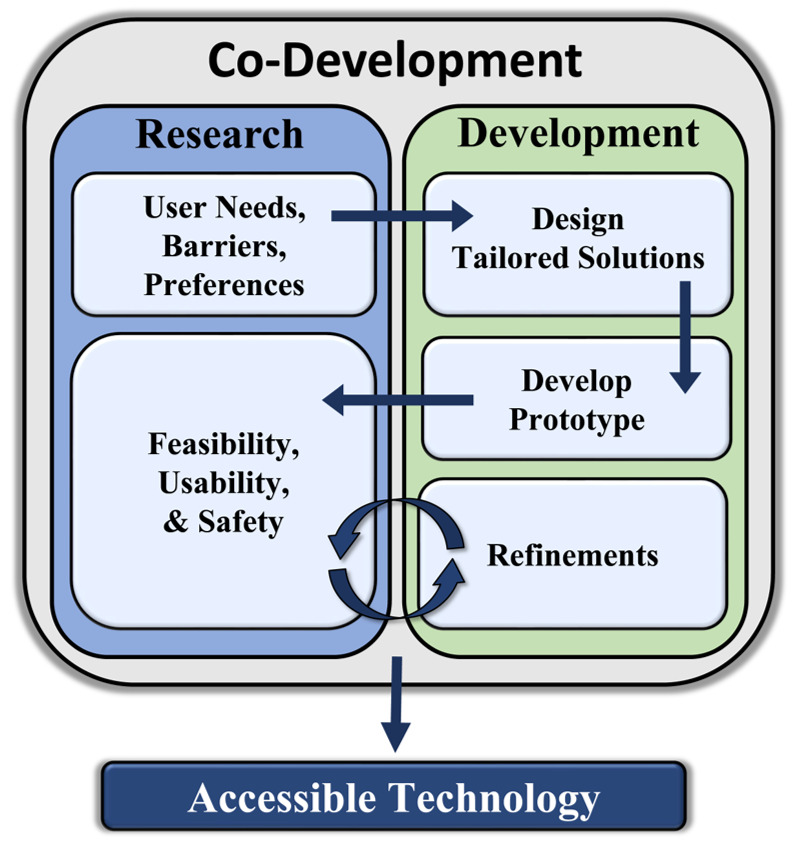
Overview of co-development process.

**Figure 3 ijerph-21-00079-f003:**
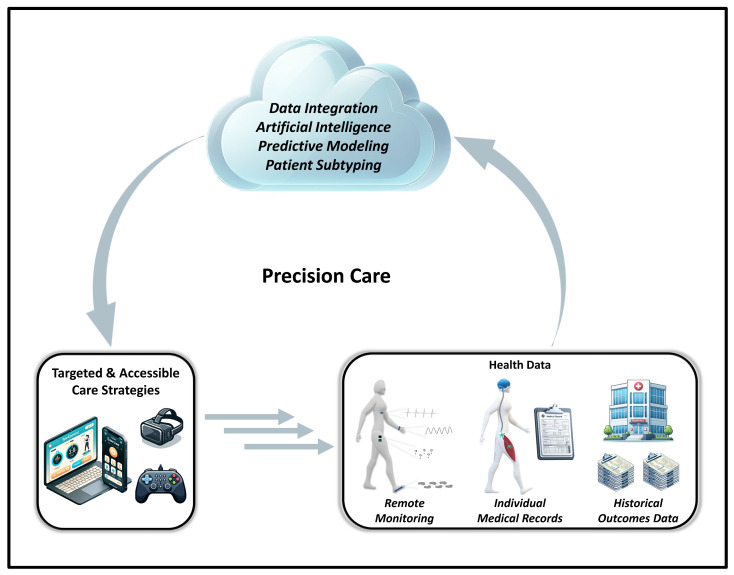
Overview of how precision rehabilitation strategies can use integrated health data to inform targeted and accessible approaches to rehabilitation and exercise, for people with disabilities.

## Data Availability

No new data were created or analyzed in this study. Data sharing is not applicable to this article.

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
