# Peer review of "Leveraging Emerging Technologies to Expand Accessibility and Improve Precision in Rehabilitation and Exercise for People with Disabilities"

_ijerph, 2024, doi:10.3390/ijerph21010079_

Round 1

Reviewer 1 Report

Comments and Suggestions for Authors

This is a well written article on a very interesting topic. Figure 1 clearly depicts the purpose of the manuscript and adds value to the paper.  

Yet, it is very extensive and could be shortened in some places, such as deleting or minimizing the text on the long history to come to the current position in technology (i.e., line 148-154). Also, for the lines 159-169 it would be better to shorten the text. The message is that online fitness is extremely popular, but in the commercial versions most of the time not useful for people living with disabilities. Line 179-182; the information in this sentence does not add added information.

Lines 445 - 466; Related to the VR and AR, it would be nice to add more concrete information about the usability of VR or AR in rehabilitation; when will it be a valid add.

Minor comments:

Line 476 has a double space between exercise and activities.

Line 568: two times the word can in this sentence. Delete the first.

Reviewer 2 Report

Comments and Suggestions for Authors

Original submission: Leveraging Emerging Technologies to Expand Accessibility and Improve Precision in Rehabilitation and Exercise for People with Disabilities.

Comments to the authors:

The authors discussed how digital health technology provides new rehabilitation and exercise opportunities for people with disabilities. Overall, the manuscript is well-written and provides a beneficial summary of utilizing digital health technology for increasing access to exercise and rehabilitation for people with disabilities. The information is not only applicable in scientific research but also in clinical practice. I have two major comments for the authors’ consideration.

Major comments:

1.     While the manuscript provides a great summary of how digital health technology can be utilized to increase access to rehabilitation and exercise among people with disabilities, the authors should further information and explain the methods used to compile this review paper.

2.     One of the notions of utilizing digital health technology is to increase access. On the other hand, using technology can be a barrier for some groups of people with and without disabilities. It would be helpful for the authors to point out and discuss these potential barriers that people with disabilities might face with digital health technology devices/services. This will also help guide future research agenda.  
